# Determinants of Diet Quality in Adolescents: Results from the Prospective Population-Based EVA-Tyrol and EVA4YOU Cohorts

**DOI:** 10.3390/nu15245140

**Published:** 2023-12-18

**Authors:** Katharina Mueller, Alex Messner, Johannes Nairz, Bernhard Winder, Anna Staudt, Katharina Stock, Nina Gande, Christoph Hochmayr, Benoît Bernar, Raimund Pechlaner, Andrea Griesmacher, Alexander E. Egger, Ralf Geiger, Ursula Kiechl-Kohlendorfer, Michael Knoflach, Sophia Zollner-Kiechl

**Affiliations:** 1VASCage, Centre on Clinical Stroke Research, Adamgasse 23, 6020 Innsbruck, Austria; katharina.mueller@student.i-med.ac.at; 2Department of Paediatrics II, Medical University of Innsbruck, Anichstrasse 35, 6020 Innsbruck, Austria; alex.messner@tirol-kliniken.at (A.M.); anna.staudt@tirol-kliniken.at (A.S.); anna.stock@i-med.ac.at (K.S.); nina.gande@tirol-kliniken.at (N.G.); christoph.hochmayr@tirol-kliniken.at (C.H.); ursula.kiechl-kohlendorfer@tirol-kliniken.at (U.K.-K.); 3Department of Paediatrics III, Medical University of Innsbruck, Anichstrasse 35, 6020 Innsbruck, Austria; johannes.nairz@tirol-kliniken.at (J.N.); ralf.geiger@tirol-kliniken.at (R.G.); 4Department of Vascular Surgery, Feldkirch Hospital, Carinagasse 41, 6800 Feldkirch, Austria; bernhard.winder@student.i-med.ac.at; 5Department of Paediatrics I, Medical University of Innsbruck, Anichstrasse 35, 6020 Innsbruck, Austria; benoit.bernar@tirol-kliniken.at; 6Department of Neurology, Medical University of Innsbruck, Anichstrasse 35, 6020 Innsbruck, Austria; raimund.pechlaner@i-med.ac.at; 7The Central Institute of Clinical Chemistry and Laboratory Medicine (ZIMCL), Medical University of Innsbruck, Anichstrasse 35, 6020 Innsbruck, Austria; andrea.griesmacher@tirol-kliniken.at; 8Institute of Hygiene and Medical Microbiology, Medical University of Innsbruck, Anichstrasse 35, 6020 Innsbruck, Austria; alexander.egger@tirol-kliniken.at; 9Department of Neurology Hochzirl Hospital, Hochzirl 1, 6170 Zirl, Austria

**Keywords:** adolescents, diet quality, AHEI-2010, DASH score, nutrition knowledge

## Abstract

(1) Background: Unhealthy dietary behaviors are estimated to be one of the leading causes of death globally and are often shaped at a young age. Here, we investigated adolescent diet quality and its predictors, including nutrition knowledge, in two large Central European cohorts. (2) Methods: In 3056 participants of the EVA-Tyrol and EVA4YOU prospective population-based cohort studies aged 14 to 19 years, diet quality was assessed using the AHEI-2010 and DASH scores, and nutrition knowledge was assessed using the questionnaire from Turconi et al. Associations were examined utilizing multivariable linear regression. (3) Results: The mean overall AHEI-2010 score was 42%, and the DASH score was 45%. Female participants (60.6%) had a significantly higher diet quality according to the AHEI-2010 and DASH score. AHEI-2010 and DASH scores were significantly associated (*p* < 0.001) with sex, school type, smoking, and total daily energy intake. The DASH score was additionally significantly associated (*p* < 0.001) with age, socioeconomic status, and physical activity. Participants with better nutrition knowledge were more likely to be older, to attend a general high school, to live in a high-income household, to be non-smokers, and to have a higher diet quality according to the AHEI-2010 and DASH score. (4) Conclusions: Predictors of better diet quality included female sex, physical activity, educational level, and nutrition knowledge. These results may aid focused interventions to improve diet quality in adolescents.

## 1. Introduction

An unhealthy diet is estimated to be one of the leading causes of death globally [1] and is associated with a higher risk of cardiovascular disease (CVD), cancer, and type 2 diabetes [2,3]. Food preferences and eating behaviors are often shaped at a young age and maintained throughout adulthood [4,5]. Traditionally, nutritional studies have evaluated individual nutrients or foods and their effect on total and cause-specific mortality [6,7,8]. However, evaluating dietary patterns may reflect the complexity of dietary intake and their relation to health outcomes more holistically [9,10,11]. Several diet quality indices, like the Alternative Healthy Eating Index 2010 (AHEI-2010) [11] and the Dietary Approaches to Stop Hypertension (DASH) score [10], have been developed to quantify adherence to recommended dietary patterns. Studies have consistently shown that many adolescents follow a poor diet [12,13,14,15]. Adolescents’ poor eating habits implicate present and future diet-related diseases [16,17,18,19]. A higher proportion of energy from fat and lower intake of micronutrients like vitamins and iron may result in nutritional inadequacies or deficits that can impair cognitive function and physical performance and lead to obesity, arterial hypertension, hypercholesterolemia, and dysglycemia [16,17]. Furthermore, maintaining healthy eating behaviors during adolescence can help prevent future chronic diseases [18,19]. Understanding the determinants of unhealthy dietary behaviors in adolescents is critical in addressing the rising concerns of adolescent nutrition and health. Known determinants of unhealthy diet among adolescents are male sex, higher total energy intake, low levels of physical activity, and low socioeconomic status [20,21,22,23]. Previous studies either aimed to evaluate individual foods and nutrients [24,25], or particular subgroups with comorbidities [26,27]. However, to our knowledge, there are few studies investigating diet quality using more comprehensive diet scores and using information on possible influencing factors such as school type and nutrition knowledge. Understanding these determinants is crucial for developing effective interventions aimed at promoting healthier eating habits among adolescents. Furthermore, few large studies were conducted in Central Europe. Therefore, the aim of this study was to examine the diet quality, according to a diet score that is associated with the risk of general chronic disease (AHEI-2010) and a second one associated with the risk of CVD (DASH score), as well as nutrition knowledge and their association with age, sex, and school type in a large community-based cohort of Central European adolescents aged 14 to 19 years. 

## 2. Materials and Methods

### 2.1. Study Population

This cohort study used data from the Early Vascular Ageing in the YOUth—EVA4YOU (NCT04598685) and the Early Vascular Ageing (EVA)-Tyrol (NCT03929692) studies, which both included healthy adolescents of the general population. The EVA4YOU study is a cross-sectional study that enrolled 1517 adolescents with a targeted age between 14 and 19 years of age from North Tyrol and was conducted between January 2021 and March 2023. The EVA-Tyrol study is a community-based, non-randomized controlled study that enrolled 2102 participants aged 14 to 22 years from high schools and companies spread across North Tyrol, East Tyrol (Austria), and South Tyrol (Italy) and was performed between May 2015 and July 2018. Although the EVA-Tyrol study administered a health intervention to a subgroup of participants, only data from the baseline examination before the health intervention were included in the current analysis. In both cohorts, components and determinants of cardiovascular health were collected at the schools‘ or companies’ sites and included a self-administered and assisted questionnaire, a structured interview, and a series of examinations (blood sampling, high-resolution ultrasound of the carotid arteries, measurement of carotid–femoral pulse-wave velocity, blood pressure measurement, and anthropometry) conducted by specially trained medical staff. The EVA-Tyrol study protocol and detailed information about data collection and measurement procedures are described elsewhere [28]. Data collection and measurement procedures of the EVA4YOU study were similar to those of the EVA-Tyrol study, except for using an electronic case report form. The study protocols of the EVA4YOU (1053/2020) and the EVA-Tyrol (AN2015-0005 345/4.13) trials were approved by the ethics committee of the Medical University Innsbruck, and the studies were conducted in accordance with the Declaration of Helsinki. Written informed consent of the participants and their legal representative, in case the participant was younger than 18 years, was obtained in both studies. 

For the present analysis, we included participants from both studies aged 14 to 19 years. We excluded participants with missing information regarding diet. As suggested in previous publications [29,30], we furthermore excluded those who had a daily energy intake of less than 600 kcal or more than 3500 kcal for women and less than 800 kcal or more than 4200 kcal for men to limit the bias of an increased energy intake allowing for the consumption of more potentially healthy food items. The flow diagram of the sample selection is shown in Figure 1. 

School types are categorized according to the Austrian school system where, after 9 years of mandatory school, students may follow three different educational pathways. They may attend secondary academic high schools with general academic education (general high schools), vocation-directed secondary schools offering general and occupation-specific knowledge (profession-guided high school), or become apprentices within specific companies and regularly visit vocational schools (vocational school). 

### 2.2. Procedures and Assessments

Study personnel performed anthropometric measurements with calibrated portable electronic scales (Soehnle style sense compact 200, Backnang, Germany) that included size and weight. Body Mass Index (BMI) was calculated as weight (in kilograms) divided by height (in meters) squared. To assess the weight status, BMI values were expressed as percentile values and Z-scores for the appropriate age and sex according to a German reference dataset [31]. According to the US Centers for Disease Control and Prevention, we classified the BMI groups as follows: underweight (<5th percentile), normal weight (≥5th and <85th percentile), overweight (≥85th and <95th percentile), and obese (≥95th percentile) [32]. 

Smoking status and amount of moderate to intense physical activity in minutes per day were reported by the participants in a face-to-face interview. Assessment of smoking status distinguished between never smoked (never smoked a whole cigarette), previous smokers (including number who practiced abstinence), and current smokers (smoked within the last 30 days). Participants were enquired about their mean moderate- and vigorous-intensity activity in minutes per day during school hours and leisure time in accordance with Americans Heart Association’s Life’s Simple 7 [33] in a face-to-face interview. Both answers were added up and considered their physical activity in minutes per day. Examples of moderate- and vigorous-intensity exercises were given according to Physical Activity Guidelines for Americans [34]. 

Socioeconomic status (SES) was determined with the Family Affluence Scale (FAS II) based on four questions related to household characteristics like owning a car, traveling on vacation with the family, having one’s own bedroom, and the family having a computer. To calculate the FAS(II), the points were summed up, ranging from 0 to 9, and divided into three categories: low (0–2 points), medium (3–5 points), and high (6–9 points) SES. For international use, the FAS(II) Scale was developed and validated within the Health-Behavior in School-aged Children (HBSC) cross-national study [35,36]. 

Dietary information was collected with the use of a validated food frequency questionnaire (FFQ) [37] on the basis of the gold-standard FFQ by Willett et al. [38]. Examples of reported used serving sizes were provided on every single food item. The assessment referred to the frequency of consumption within the last month. The frequency of consumption of each food item was determined by using 9 different frequency categories ranging from never or less than once a month to six or more times a day. Energy and nutrient intakes for each participant were calculated by multiplying the frequency of consumption of each food by its energy or nutrient content and summing nutrient contributors across all of the food items. Energy and nutrient intake were calculated by converting reported food intake into macronutrient intake values for each food using the reported serving size and the United States Department of Agriculture’s (USDA) Food and Nutrient Database for Dietary Studies [39].

Diet quality was estimated by using two scores, the AHEI-2010 and the DASH score. The AHEI-2010 is a diet quality index based on foods or food components associated with chronic disease risk [11,40]. Based on the Dietary Guidelines for Americans and as used in numerous studies investigating healthy eating patterns [11,29,30], in this study, the AHEI-2010 is composed of ten components: whole fruit, vegetable (excluding potatoes), whole grains, red and processed meat, nuts and legumes, long-chain (ω-3) fats (eicosapentaenoic acids (EPA) and docosahexaenoic acids (DHA)), polyunsaturated fatty acids, trans-fat, sugar-sweetened beverages, and sodium. The score of each component ranges from 0 (worst) to 10 (best), with a maximum score of 100 (highest adherence). Alcohol intake was not included as a component of the AHEI-2010 score in our study due to inconsistent intake across different age groups and strong spirits not being legal until the age of 18 years in Austria and Italy. Detailed assessment of the AHEI-2010 score components and criteria for scoring can be found in Appendix A. Adherence to DASH diet has been shown to have a protective effect against the incidence of CVD [10,41]. The DASH diet emphasizes the consumption of whole grains, fruits and vegetables, low-fat dairy, lean meat, fish, poultry, nuts, seeds, and legumes, and sparse use of fats and oils. The DASH score created by Fung et al. [10] consists of eight components: fruit, vegetables, nuts and legumes, whole grains, low-fat dairy products, sodium intake, red and processed meat, and sugar-sweetened beverages. For the first five components, participants were given one (lowest quintile) to five (highest quintile) points. Scores were reversed (five to one for the lowest to highest quintile) for sodium intake, red and processed meat, and the consumption of sugar-sweetened beverages. The DASH-Diet score ranged from eight (minimal adherence) to 40 (maximal adherence). Detailed assessment of the DASH score components and criteria for scoring can be found in Appendix A. 

Nutritional knowledge was only assessed within the EVA-Tyrol study group using a self-administered questionnaire containing two sections (E, H) of a dietary questionnaire originally developed and validated by Turconi et al. [42]. Section E aimed to investigate the student’s beliefs about healthy and unhealthy diets and food. It consists of five questions, each having four different responses, with a score ranging from one to four and a maximum score of 20. Section H contains eleven questions focusing on nutritional knowledge. The correct response receives one point, summing up to a maximum score of eleven points. The remaining 8 sections originally included in the Turconi dietary questionnaire were designed to assess personal data (A), food habits (B, C), eating behavior, physical activity and lifestyle (D, F, G), and food safety (I, J) and were either not relevant to our study design or otherwise assessed and therefore not recorded. 

### 2.3. Statistical Analysis

For the present study, characteristics are shown as mean ± standard deviation (mean ± SD) if normally distributed and otherwise expressed by median and interquartile range (M (Q1, Q3)). Count data were described as n (%). Differences between groups were analyzed using the *t*-test, Mann–Whitney-U, or Chi-square-Test. The diet quality scores (AHEI-2010 and DASH score) were analyzed separately in quartiles based on the distribution of each score. The comparison of values across different quartiles was conducted by using a univariate ANOVA or Pearson Chi-square Test. A multivariable linear regression model with adjustment for age, sex, school type, socioeconomic status, BMI-Z Score, and energy intake was used to examine the association of adolescent factors with each diet quality score. Furthermore, a linear regression analysis on nutritional knowledge was performed for the EVA-Tyrol study, in which these data were available. *p* values were considered statistically significant at *p* < 0.05. Diet scores were calculated using Microsoft Excel 2016 (Microsoft Inc., Redmond, WA, USA) and SPSS version 29.0 (SPSS Inc., Chicago, IL, USA). All statistical analyses were conducted using SPSS version 29.0 (SPSS Inc., Chicago, IL, USA) and R version 4.3.1 for Windows (R Foundation for Statistical Computing, Vienna, Austria). 

## 3. Results

### 3.1. Baseline Characteristics

As shown in Table 1, the total study sample of 3056 participants included 1329 (43.5%) from the EVA4YOU study and 1727 (56.5%) from the EVA-Tyrol study. In the total sample, the mean age was 16.8 years, 60.6% were female, the majority (54.5%) attended a profession-guided high school, and the daily median energy intake according to the FFQ was 1975 kcal. Dietary patterns were described using the AHEI-2010 and the DASH score. The median AHEI-2010 score was 42 points (range 12–82 points), and the median DASH score was 18 points (range of 8–32 points). Participants of the EVA4YOU study tended to be older, more often female, more often living in high-income households, were less often obese, more physically active, and scored fewer points on the AHEI-2010 score and more on the DASH score in comparison to participants of the EVA-Tyrol study.

### 3.2. Distribution of Characteristics across Quartiles of the AHEI-2010 Score 

The distribution of the study characteristics across quartiles of the AHEI-2010 score among study participants is shown in Table 2. Participants with a higher AHEI-2010 score were more likely to be female, non-smokers, and to have a higher energy intake. A higher AHEI-2010 score was associated with a higher intake of every single healthy food item included in the AHEI-2010 score. Age, FAS-II Score, BMI Z-score, and amount of physical activity did not significantly differ between the AHEI-2010 quartiles. Adolescents scored best in having a high intake of long-chain (ω-3) fats (EPA + DHA) and nuts and legumes, as well as a low intake of trans fat. However, only 27% did not exceed the recommended maximum daily sodium intake, and only 8% ate five or more portions of vegetables daily (Appendix A). 

### 3.3. Distribution of Characteristics across Quartiles of the DASH Score

The distribution of the study characteristics across quartiles of the DASH score among study participants is shown in Table 3. Participants in the highest DASH score quartiles were more likely to be older, to be female, to attend a general high school, to live in a high-income household according to their FAS(II) Score, to be more physically active, to smoke, and to consume more kcal per day. There were no significant differences in BMI Z-score across the quartiles of DASH score. Furthermore, a higher DASH score quartile was associated with a more frequent intake of all healthy food items, especially fruit, vegetables, nuts and legumes, and whole grains. Adolescents scored best in consuming 4 or more portions of fruit (20.1%), less than 2.8 portions of red and processed meat a week (19.1%), and no sugar-sweetened beverages daily (15.8%). The DASH score item with the lowest score was low-fat dairy products, which were consumed daily by only 25% of study participants, and within this group, only 3% consumed the recommended 2.3 portions daily (Appendix A). 

### 3.4. Association of Adolescent Factors and the Dietary Scores AHEI-2010 and DASH Score

The relationship between two nutrition scores, the AHEI-2010 and the DASH score, and adolescent factors are displayed in Table 4. In a sex- and age-adjusted linear regression, attending vocational school was significantly associated with a lower AHEI-2010 score, and attending general high school, being physically active, being a never-smoker, and a higher total energy intake were associated with a higher AHEI-2010 score. In a univariate linear regression model, female sex was significantly associated with a higher AHEI-2010 score. A further multivariable analysis indicated that attendance at a general high school, a higher socioeconomic status, more physical activity, and a higher total energy intake were associated with a higher DASH score. In a univariate linear regression model, older age and the female sex were significantly associated with a higher DASH score.

As shown in Figure 2, female sex, attending general high school, BMI Z-score, physical activity, smoking, and total energy intake remained significantly associated with higher AHEI-2010 scores after multivariable adjustment. Furthermore, after multivariable adjustment, a higher DASH score was significantly associated with older age, female sex, attending general high school, FAS(II) Score, more frequent physical activity, smoking, and a higher total energy intake. Sensitivity analysis was conducted adjusting for the study with similar results (not shown).

### 3.5. Nutrition Knowledge

We also assessed the distribution of adolescent factors based on nutritional knowledge, which was only available in the EVA-Tyrol population. The results of each section were described separately and are summarized in Table 5. Participants with a higher score in Section E, aiming at investigating the student’s beliefs about healthy and unhealthy diet and food, were more likely to be female and to attend a general high school. Students with a higher score in Section H, evaluating nutritional knowledge, were significantly more likely to attend a general high school and to have a higher diet quality according to the AHEI-2010 and DASH scores. 

We also assessed the association between diet quality and nutritional knowledge. The results are displayed in Table 6. EVA-Tyrol participants attending a general high school, having a higher BMI, and not smoking have a better understanding of a healthy diet after adjusting for sex and age. Participants with better nutrition knowledge were more likely to be older, to attend a general high school, to live in a high-income household, to not smoke, and to have a higher diet quality according to the AHEI-2010 and DASH scores. There was no significant association with female sex or physical activity.

## 4. Discussion

Using data from two large community-based cohort studies of adolescents—the EVA4YOU and the EVA-Tyrol study—this analysis examined the prevalence of diet quality and knowledge about healthy diets depending on age, sex, and school type. The overall diet quality of the study participants using the AHEI-2010 and DASH scores was low in both sexes and across all ages and education types. This was the result of low scores in all diet components, but especially of a high intake of sodium and red and processed meat and a low intake of polyunsaturated fatty acids. The results indicate that higher AHEI-2010 scores were associated with female sex and attending general high school, as well as increased physical activity, non-smoking status, and a higher total energy intake. Additionally, participants with greater knowledge about nutrition had higher AHEI-2010 scores, whereas beliefs about healthy and unhealthy diets and foods showed no such association. Furthermore, a positive relationship between higher DASH scores and more frequent physical activity, non-smoking status, and a higher total energy intake, as well as greater nutrition knowledge, was observed. 

Our results are generally consistent with previous studies that have separately investigated diet quality scores [43,44,45,46,47,48,49] and the prevalence of knowledge about healthy diets [50,51] in adolescents. Within this sample of Tyrolean adolescents, the average AHEI-2010 score was 42 points, which suggests 42% adherence to recommendations to achieve an optimal diet quality and is therefore rather low. However, similar diet quality index scores were determined in other youth populations. A cross-sectional study by Zheng et al. described a US-population of 5934 adolescents aged 12 to 19 years with a mean AHEI-2010 score of 29.17. However, in this study, the AHEI-2010 score was composed of 9 components and a maximal score of 90, missing alcohol intake and long-chain (ω-3) fats, resulting in a 32% healthy diet adherence [43]. Ducharme-Smith et al. showed a mean AHEI-2010 score of 47.4 in a sample size including 240 Native American adolescents with diabetes or prediabetes with a mean age of 13.6 years [44]. In contrast to our analysis, alcohol intake was reflected in the AHEI-2010 score, resulting in 11 components and a possible count of 110. Therefore, a 43% adherence to dietary guidelines was found. Similar to our results, which showed a mean calorie intake of 1975 kcal, a mean energy intake of 2016 kcal was found. However, Ducharme-Smith et al. described a BMI-Z-Score of 2.19, reflecting an obese study population. Wang et al. applied the same AHEI-2010 score as used in our study to assess global dietary quality, showing a mean AHEI-2010 score of 45.5 in an Austrian population aged 25 and older in 2017 [45]. Regarding the DASH diet, the participants in our study had a mean score of 18.24 points, resulting in a 45% adherence to the DASH diet. In comparison to other studies [46,47,48,49], this score seems rather low. One explanation might be the consumption of dairy products. In the population described, only 25% of participants stated that they consume low-fat dairy products. Beyond that, the FFQ only distinguished between low- and high-fat milk and did not include other low-fat dairy products. Furthermore, the mean sodium intake was 5955 mg, and only 10% of our population did not exceed the recommended maximum daily sodium intake of 2676 mg. Compared to other adolescent cohorts [44,52], this represents comparatively high amounts of daily sodium consumption. However, a study by Hasenegger et al. found a mean dietary salt intake in Austrian adults of 5.6 g daily [53], and similar findings among adults were published in other European countries [54]. 

Furthermore, the association of higher AHEI-2010 and DASH scores with female sex, increased physical activity, non-smoking status, and a higher energy intake are in line with previous findings among adults [55,56,57,58]. It is not surprising that the associations with the two diet quality scores are similar, as they are composed of several similar components such as fruits, vegetables, whole grains, nuts, and legumes. One explanation for better diet quality among female adolescents might be their higher intake of healthy food items like fruits and vegetables. Similar results were found in previous studies [59,60]. Furthermore, in our study, more female adolescents (35%) attended a general high school than male participants (25%), possibly explaining better diet quality. However, female adolescents were less physically active in min/day (median 40 min/day (IQR 25, 60)) than male participants (60 (35, 90)). No significant differences between female and male adolescents depending on age, SES, or nutrition knowledge were found in our study cohort. 

Additionally, healthy behaviors like physical activity, non-smoking, and a better diet are known to co-occur [21,61]. Meeting the recommendations on physical activity is found to be one of the main determinants for adherence to healthy dietary patterns [62]. While higher DASH scores were associated with older age, AHEI-2010 scores were not. Previous studies have found mixed results [63,64,65]. In our study, older participants tended to have a higher socio-economic status, which might play a role. 

We described an association between higher AHEI-2010 and DASH scores in adolescents attending general high school and lower scores in those attending vocational or profession-guided schools. One possible explanation is that in profession-guided high schools and vocational schools, nutritional education is an even lower priority than in general high schools. To the best of our knowledge, there were no previous studies that compared dietary patterns measured by the AHEI-2010 and DASH scores in different school types among adolescents. 

SES was associated with the DASH score. The association between SES and dietary quality has been shown previously by French et al. [66]. A low SES, on the other hand, is known to be associated with CVD risk [67], and worse dietary habits might be one of the reasons for this link. As the connection between SES and dietary quality already applies in adolescence, a timely consolidation of favorable dietary habits might be beneficial for long-term disease risk.

Although lower diet quality among adolescents is known from several studies [12,13,14,15], implicating present and future diet-related diseases like micronutrient deficiencies, obesity, or diabetes [16,17,18,19], our results help to identify subgroups that are particularly in need of timely interventions and to shape interventions programs focusing on different dietary components and including education programs. Previous studies evaluated intervention programs focusing on micronutrient supplementation and nutrition education to prevent deficiencies or obesity among adolescents [68,69,70]. However, the benefits of these interventions vary and are to be interpreted with caution [71]. On the other hand, some studies suggest that a combination of interventions, including nutrition, physical activity, and nutrition knowledge, have a greater effect than interventions focusing on them individually [71,72]. Promoting healthy nutrition behaviors among adolescents might be most successful when including a wide range of policies that are viable and implemented both in schools and at home [71,73,74]. Implementing nutrition interventions among subgroups of adolescents with lower diet quality could catalyze a shift in public health outcomes. Given the local societal context, these interventions could address specific cultural and socioeconomic factors influencing dietary habits. Particularly, subgroups with known low diet quality, such as adolescents with male sex, low SES, and low educational level, need to be a special focus in intervention programs. 

The association of better nutrition knowledge with older age, living in a high-income household, non-smoking status, and the consumption of healthy foods was previously described in several studies [50,51,75,76,77]. However, to the best of our knowledge, there were no previous studies relating AHEI-2010 or DASH scores measuring healthy dietary patterns and their association with nutrition knowledge.

The strengths of the present study include a large, well-characterized community-based sample representative of the healthy adolescent Central European population, which enabled us to associate diet quality with different school types and nutrition knowledge. A large amount of data on dietary habits were assessed, applying a previously validated FFQ to allow for an in-depth analysis. Calculating diet quality scores represents eating behavior more holistically than looking at food items individually [9,10,11]. Furthermore, foods and nutrients identified in the AHEI-2010 score have been consistently associated with a lower risk of general chronic disease [10]. Components of the DASH score are based on the DASH-style diet, and the DASH score was found to lower blood pressure and the risk of chronic heart disease and stroke [11]. To our knowledge, this study is the first to examine the association of two different dietary scores with school type and nutrition knowledge in a large group of healthy Central European adolescents. 

However, this study has certain limitations. First, measurement error may hamper self-reported dietary behavior. However, for the FFQ employed here, validation using short-term recall has been performed [37], and good agreement with dietary assessment using the FFQ developed by Willet et al. in multiple cohorts has been shown [78,79,80,81]. Second, as with most FFQs, we did not cover discretionary salt intake in cooking or seasoning. Therefore, the sodium component in both the AHEI-2010 and the DASH score must be interpreted cautiously, and an underestimation might have occurred. Third, we excluded information about daily alcohol intake in calculating the AHEI-2010 score due to inconsistent intake across different age groups and strong spirits not being legal until the age of 18. However, the same method was used in several previously mentioned cohort studies. Fourth, we calculated the FAS(II) Scale instead of the most recent FAS(III) Scale better reflecting SES in high-income countries like Austria. However, data from the EVA-Tyrol study were obtained before the updated Family Affluence Scale was published. Fifth, this study lacked information on parental and family eating habits, which may be a confounding factor, especially at the mean age of 16 years. Finally, as in all observational studies, we cannot exclude the possibility of residual and unmeasured confounding. More prospective studies are needed to determine the relationship between different diet quality scores and the positive effect of extensive nutrition knowledge on healthy eating patterns in adolescents.

## 5. Conclusions

In this study, higher AHEI-2010 and DASH scores were associated with female sex, higher levels of physical activity, non-smoking status, and attendance at a profession-guided or general high school among adolescents. In addition, higher DASH scores were associated with older age and living in a high-income household. Furthermore, greater knowledge about nutrition resulted in a healthier diet measured by AHEI-2010 and DASH scores. However, adolescent’s beliefs about healthy and unhealthy diets and food were not reflected in either diet quality score. Hence, to improve dietary quality, which was, in general, low in all sub-groups of adolescents, putting more emphasis on nutrition education might be beneficial. Furthermore, a disadvantage faced by adolescents with low SES and lower education levels has been found. These groups are therefore in particular need of interventions and improving nutritional education.

## Figures and Tables

**Figure 1 nutrients-15-05140-f001:**
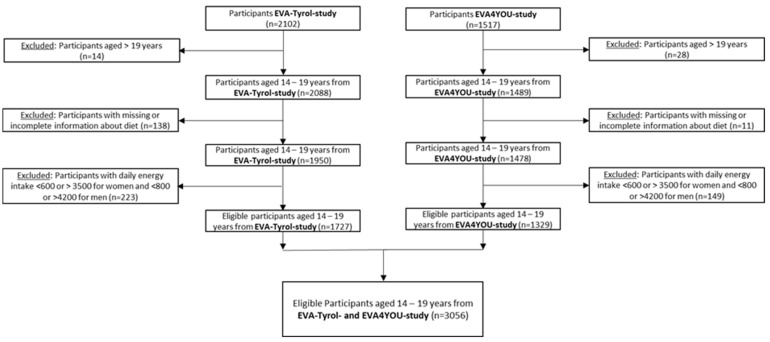
Consort flow diagram of the sample selection.

**Figure 2 nutrients-15-05140-f002:**
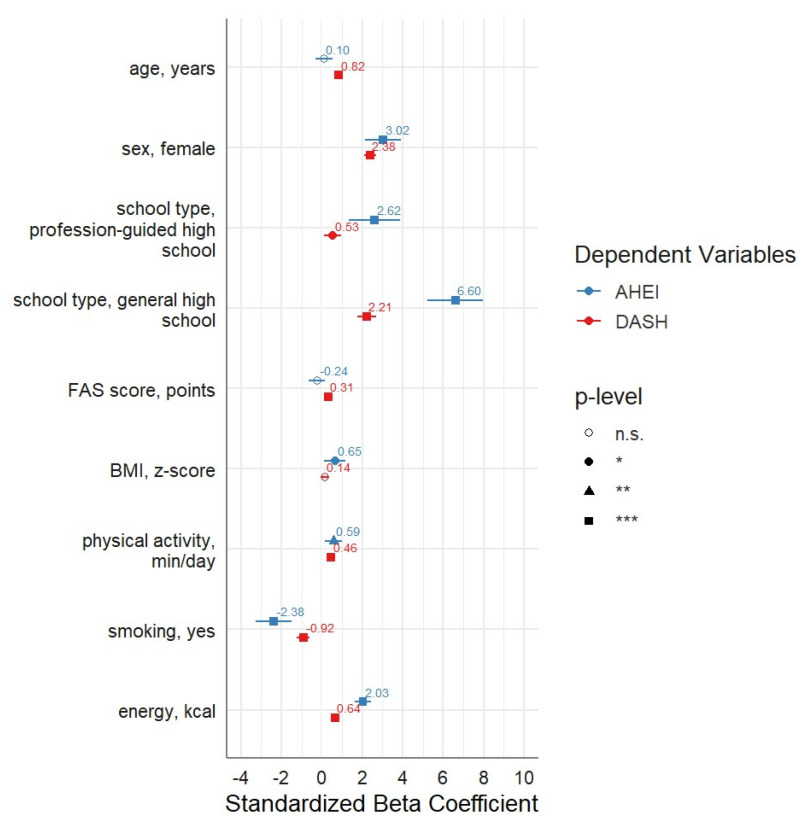
Independent influence of adolescent factors on nutrition scores. * <0.05, ** <0.01, *** <0.001. Multivariable linear regression on AHEI-2010 and DASH scores. Adjustment for age, sex, school type, socioeconomic status, BMI Z-score, physical activity, smoking status, and energy intake.

**Table 1 nutrients-15-05140-t001:** Descriptive characteristics of participants in EVA4YOU and EVA-Tyrol studies.

Variable	Total Sample (*n* = 3056)	EVA4YOU (*n* = 1329)	EVA-Tyrol (*n* = 1727)	*p*
Age, years, mean ± SD	16.8 ± 1.3	17.2 ± 1.3	16.5 ± 1.2	<0.001 *
Sex, *n* (%)				<0.001 ***
male	1205 (39.4)	468 (35.2)	737 (42.7)	
female	1851 (60.6)	861 (64.8)	990 (57.3)	
Schooltype, *n* (%)				<0.001 ***
Vocational school	437 (14.3)	264 (19.9)	173 (10.0)	
Profession-guided high school	1667 (54.5)	678 (51.0)	989 (57.3)	
General high school	952 (31.2)	387 (29.1)	565 (32.7)	
FAS(II) Scale, *n* (%) ^a^				<0.001 ***
Low	18 (0.6)	4 (0.3)	14 (0.8)	
Medium	821 (26.9)	262 (19.7)	559 (32.4)	
High	2192 (71.7)	1063 (80.0)	1129 (66.3)	
BMI, kg/m^2^, mean ± SD	22.06 ± 3.63	22.29 ± 3.77	21.89 ± 3.51	0.003 *
BMI Z-score, mean ± SD	0.11 ± 1.04	−0.05 ± 1.04	0.22 ± 1.02	<0.001 *
Weight Group, *n* (%) ^b^				<0.001 ***
Underweight	133 (4.4)	84 (6.3)	49 (2.8)	
Normal	2347 (76.8)	1039 (78.2)	1308 (75.7)	
Overweight	347 (11.4)	148 (11.1)	199 (11.5)	
Obesity	229 (7.5)	58 (4.4)	171 (9.9)	
Physical activity, min/d, M (Q1, Q3)	45 (30, 75)	55 (30, 80)	45 (25, 60)	<0.001 **
Total Energy, kcal/d, M (Q1, Q3)	1975 (1483, 2542)	2014 (1548, 2586)	1941 (1439, 2517)	0.004 **
Never-smoker, *n* (%)	2059 (67.4)	843 (63.4)	1216 (70.4)	<0.001 ***
AHEI-2010 score, M (Q1, Q3)	42 (34, 50)	39 (32, 47)	44 (36, 52)	<0.001 **
DASH score, M (Q1, Q3)	18 (15, 21)	19 (17, 22)	17 (14, 20)	<0.001 **

FAS(II) Scale (Family Affluence Scale II). BMI (body mass index). * *t*-test. ** ANOVA. *** Pearson Chi-squared Test. ^a^ Low (0–2 points), medium (3–5 points), and high (6–9 points). ^b^ Underweight (<5th percentile), normal weight (≥5th to <85th percentile), overweight (≥85th to <95th percentile), and obese (≥95th percentile).

**Table 2 nutrients-15-05140-t002:** Distribution of study characteristics across quartiles of the AHEI-2010 score among EVA4YOU and EVA-Tyrol participants.

	Quartiles of AHEI-2010 Score				
Characteristics	Q1 (*n* = 805)	Q2 (*n* = 777)	Q3 (*n* = 745)	Q4 (*n* = 729)	*p*
AHEI-2010 score, M (Q1, Q3) Possible Range of 0–100	29 (25, 32)	39 (36, 40)	46 (44, 48)	57 (53, 61)	<0.001 *
Age, years, mean ± SD	16.9 ± 1.3	16.8 ± 1.3	16.7 ± 1.3	16.8 ± 1.3	0.200 *
Female, *n* (%)	461 (57.3)	445 (57.3)	449 (60.3)	496 (68.0)	<0.001 **
Schoolt ype, *n* (%)					<0.001 **
Vocational school	171 (21.2)	102 (13.1)	96 (12.9)	68 (9.3)	
Profession-guided high school	468 (58.2)	438 (56.4)	423 (56.8)	338 (46.4)	
General high school	166 (20.6)	237 (30.5)	226 (30.3)	323 (44.3)	
FAS(II) Score, Points, M (Q1, Q3)	6 (5, 7)	6 (6, 8)	6 (5, 8)	6 (5, 8)	0.357 *
BMI Z-score, M (Q1, Q3)	0.03 (−0.64, 0.74)	0.14 (−0.60, 0,77)	0.11 (−0.60, 0.84)	0.12 (−0.54, 0.80)	0.192 *
Physical activity, min/d, M (Q1, Q3)	45 (25, 70)	45 (30, 75)	45 (30, 70)	60 (30, 80)	0.435 *
Never-smoker, *n* (%)	491 (61.0)	532 (68.5)	513 (68.9)	523 (71.7)	<0.001 **
Total energy, kcal/d, M (Q1, Q3)	1857 (1401, 2330)	1909 (1466, 2423)	2070 (1526, 2653)	2141 (1613, 2762)	<0.001 *
Components of AHEI-2010, M (Q1, Q3)					
Fruits ^a^, servings/d	1.22 (0.70, 1.91)	1.65 (1.05, 2.56)	2.49 (1.56, 3.71)	3.52 (2.21, 5.11)	<0.001 *
Vegetables ^b^, servings/d	1.29 (0.71, 1.94)	1.91 (1.27, 2.64)	2.47 (1.76, 3.43)	3.56 (2.41, 4.99)	<0.001 *
Whole grain, g/d	23.65 (3.63, 55.00)	27.5 (7.70, 55.00)	43.45 (15.12, 55.00)	43.45 (23.65, 55.00)	<0.001 *
Red and processed meat intake ^c^, servings/d	3.05 (1.81, 5.19)	2.62 (1.39, 4.84)	1.90 (0.69, 3.61)	0.89 (0.26, 2.40)	<0.001 *
Nuts and legumes ^d^, servings/d	0.14 (0.07, 0.35)	0.35 (0.14, 0.71)	0.64 (0.29, 1.12)	1.20 (0.72, 1.97)	<0.001 *
Long-chain (ω-3) fats (EPA + DHA), mg/d	0 (0, 40)	90 (0, 790)	130 (2, 830)	640 (90, 920)	<0.001 *
Polyunsaturated fatty acids, % of energy	2.35 (1.70, 4.44)	3.01 (1.82, 4.85)	3.72 (2.01, 5.13)	4.27 (2.33, 5.48)	<0.001 *
Trans fat, % of energy	0.67 (0.58, 0.78)	0.64 (0.54, 0.74)	0.60 (0.50, 0.69)	0.51 (0.42, 0.63)	<0.001 *
SSBs and fruit juice ^e^, servings/d	1.20 (0.62, 2.11)	0.88 (0.42, 1.82)	0.84 (0.42, 1.64)	0.42 (0.14, 0.91)	<0.001 *
Sodium intake, mg/d	5920 (4420, 7835)	5540 (3980, 7820)	5480 (3585, 7705)	4800 (3220, 6740)	<0.001 *

* ANOVA. ** Pearson Chi-squared Test. EPA (eicosapentaenoic acids). DHA (docosahexaenoic acids). SSBs (sugar-sweetened beverages). ^a^ 1 serving = medium piece of fruit, ^b^ 1 serving = 0.5 cup of typical local vegetables, except potatoes, ^c^ 1 serving = 113.4 g of unprocessed and 42.5 g of processed meat, ^d^ 1 serving = 28.35 g, ^e^ 1 serving = 236.6 mL.

**Table 3 nutrients-15-05140-t003:** Distribution of study characteristics across quartiles of the DASH score among EVA4YOU and EVA-Tyrol participants.

	Quartiles of DASH Score				
Characteristics	Q1 (*n* = 819)	Q2 (*n* = 851)	Q3 (*n* = 722)	Q4 (*n* = 664)	*p*
DASH score, M (Q1, Q3) Possible Range of 8–40	14 (12, 15)	17 (16, 18)	20 (19, 21)	24 (22, 26)	<0.001 *
Age, years, mean ± SD	16.5 ± 1.3	16.8 ± 1.3	16.9 ± 1.3	17.0 ± 1.3	<0.001 *
Female, *n* (%)	374 (45.3)	503 (59.1)	476 (66.2)	498 (75.3)	<0.001 **
School type, *n* (%)					<0.001 **
Vocational school	154 (18.8)	131 (15.4)	83 (11.5)	69 (10.4)	
Profession-guided high school	510 (62.3)	483 (56.8)	386 (53.5)	288 (43.4)	
General high school	155 (18.9)	237 (27.8)	253 (35.0)	307 (46.2)	
FAS(II) Score, Points, M (Q1, Q3)	6 (5, 7)	6 (5, 8)	7 (5, 8)	7 (6, 8)	<0.001 *
BMI Z-score, M (Q1, Q3)	0.10 (−0.59, 0.78)	0.12 (−0.58, 0.81)	0.08 (−0.63, 0.82)	0.09 (−0.57, 0.78)	0.174 *
Physical activity, min/d, M (Q1, Q3)	45 (25, 75)	45 (30, 70)	50 (30, 75)	60 (30, 80)	<0.001 *
Never-smoker, *n* (%)	522 (63.7)	566 (66.5)	499 (69.1)	472 (71.1)	0.016
Total energy, kcal/d, M (Q1, Q3)	1921 (1421, 2488)	1898 (1464, 2407)	1986 (1463, 2571)	2163 (1631, 2682)	<0.001 *
Components of DASH score, M (Q1, Q3)					
Fruit and fruit juice ^a^, servings/d	1.35 (0.77, 2.00)	1.91 (1.19, 3.07)	2.70 (1.63, 4.09)	3.70 (2.43, 5.59)	<0.001 *
Vegetables and vegetable juice ^b^, servings/d	1.35 (0.78, 2.00)	1.91 (1.26, 2.64)	2.51 (1.64, 3.48)	3.73 (2.56, 5.00)	<0.001 *
Nuts and legumes ^c^, servings/d	0.25 (0.12, 0.58)	0.35 (0.14, 0.72)	0.45 (0.21, 0.95)	0.85 (0.43, 1.57)	<0.001 *
Whole grain ^d^, servings/d	0.43 (0.14, 0.79)	0.43 (0.14, 1.00)	0.79 (0.43, 1.00)	1.00 (0.43, 1.75)	<0.001 *
Low-fat dairy products ^e^, servings/d	0 (0, 0)	0 (0, 0)	0 (0, 0.07)	0 (0, 0.10)	<0.001 *
Sodium intake, mg/d	6170 (4455, 8480)	5510 (4050, 7630)	5230 (3510, 7330)	4770 (3300, 6550)	<0.001 *
Red and processed meat intake ^f^, servings/d	1.93 (1.2, 2.87)	1.18 (0.68, 2.14)	0.85 (0.42, 1.57)	0.46 (0.14, 0.97)	<0.001 *
SSBs ^g^, servings/d	1.44 (0.56, 2.94)	0.69 (0.26, 1.72)	0.33 (0.04, 1.12)	0.14 (0, 0.56)	<0.001 *

* ANOVA. ** Pearson Chi-squared Test. ^a^ 1 serving = medium piece of fruit or ½ cup of fruit juice, ^b^ 1 serving = 0.5 cup of typical local vegetables or ½ cup of vegetable juice, ^c^ 1 serving = 1/3 cup, 42 g, ^d^ 1 serving = 1 slice of bread, ½ cup of cooked grains, ^e^ 1 serving = 1 cup, 250 mL, ^f^ 1 serving = 30 g of red or processed meat, ^g^ 1 serving = 1 cup, 250 mL.

**Table 4 nutrients-15-05140-t004:** Association between adolescent factors and nutrition scores.

	AHEI-2010 Score			DASH Score		
	Regression Coefficient (95% CI)	R^2^	*p*	Regression Coefficient (95% CI)	R^2^	*p*
Age, years *	−0.250 (−0.570–0.071)	0.001	0.126	0.524 (0.410–0.638)	0.026	<0.001
Sex, female *	2.292 (1.44–3.139)	0.009	<0.001	1.982 (1.683–2.281)	0.052	<0.001
School type **						
Profession-guided high school	−2.124 (−2.954–−1.293)	0.018	<0.001	−1.098 (−1.385–−0.810)	0.096	<0.001
General high school	4.575 (3.689–5.460)	0.042	<0.001	1.896 (1.590–2.202)	0.122	<0.001
FAS(II) Score, Points	0.226 (−0.044–0.496)	0.011	0.101	0.326 (0.233–0.419)	0.094	<0.001
BMI Z-score	0.331 (−0.069–0.731)	0.011	0.105	−0.067 (−0.207–0.072)	0.080	0.343
Physical activity, min/d	0.010 (0.002–0.018)	0.012	0.012	0.008 (0.006–0.011)	0.092	<0.001
Smoking, never	2.582 (1.680–3.484)	0.020	<0.001	1.013 (0.699–1.326)	0.091	<0.001
Total energy, kcal/d	0.003 (0.002–0.003)	0.038	<0.001	0.001 (0.001–0.001)	0.107	<0.001

Linear regression with adjustment for sex and age (if not otherwise specified). * Analyzed with univariate linear regression. ** Reference Vocational School. CI, Confidence Interval. BMI, Body Mass Index. FAS(II) Score, Family Affluence Score II.

**Table 5 nutrients-15-05140-t005:** Distribution of beliefs about healthy and unhealthy diet and food (Section E) and nutrition knowledge (Section H) according to the study characteristics: EVA-Tyrol population (*n* = 1727).

	Nutrition Knowledge Score			
Variable	Section E Possible Range: 5–20	*p* *	Section H Possible Range: 0–11	*p* *
Sex		0.034		0.483
Male	12 ± 2		6 ± 2	
Female	12 ± 1		6 ± 2	
Schooltype		<0.001		<0.001
Vocational school	11 ± 2		5 ± 2	
Profession-guided high school	12 ± 1		6 ± 2	
General high school	12 ± 1		7 ± 2	
FAS(II) Scale ^a^,		0.276		0.628
Low	11 ± 2		6 ± 2	
Medium	12 ± 2		6 ± 2	
High	12 ± 1		6 ± 2	
Weight Group ^b^		0.34		0.615
Underweight	12 ± 2		6 ± 2	
Normal	12 ± 2		6 ± 2	
Overweight	12 ± 2		7 ± 2	
Obesity	12 ± 2		7 ± 2	
AHEI-2010 score Quartiles		0.225		0.003
Quartile 1	12 ± 1		6 ± 2	
Quartile 2	12 ± 2		6 ± 2	
Quartile 3	12 ± 1		6 ± 2	
Quartile 4	12 ± 2		7 ± 2	
DASH score Quartiles		0.412		<0.001
Quartile 1	12 ± 2		6 ± 2	
Quartile 2	12 ± 1		6 ± 2	
Quartile 3	12 ± 1		7 ± 2	
Quartile 4	12 ± 2		7 ± 2	

Values are shown as mean ± SD. * Pearson Chi-squared Test. ^a^ Low (0–2 points), medium (3–5 points), high (6–9 points). ^b^ Underweight (<5th percentile), normal weight (≥5th to <85th percentile), overweight (≥85th to <95th percentile), and obese (≥95th percentile).

**Table 6 nutrients-15-05140-t006:** Association of beliefs about healthy and unhealthy diet and food (Section E) and nutrition knowledge (Section H) and adolescent factors: EVA-Tyrol population (*n* = 1727).

	Section E			Section H		
	Standardized Beta Coefficient	R^2^	*p*	Standardized Beta Coefficient	R^2^	*p*
Age, years *	0.041	0.002	0.107	0.255	0.065	<0.001
Sex, female *	0.021	0.000	0.423	−0.011	0.000	0.664
School type **						
Profession-guided high school	0.001	0.002	0.966	−0.095	0.074	<0.001
General high school	0.071	0.007	0.006	0.187	0.099	<0.001
FAS(II) Scale	0.049	0.005	0.061	0.085	0.074	<0.001
BMI Z-score	−0.064	0.006	0.014	0.016	0.064	0.540
Physical activity, min/d	−0.018	0.003	0.493	−0.012	0.066	0.638
Smoking, never	0.075	0.008	0.005	0.101	0.075	<0.001
AHEI-2010 score	0.000	0.002	0.990	0.166	0.092	<0.001
DASH score	0.003	0.002	0.908	0.181	0.095	<0.001

Linear regression with adjustment for sex and age (if not otherwise specified). * Analyzed with univariate linear regression. ** Reference Vocational School.

## Data Availability

The data that support the findings of the study are available on request from the corresponding authors after establishing an appropriate data transfer agreement.

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
