# Peer review of "Determinants of Diet Quality in Adolescents: Results from the Prospective Population-Based EVA-Tyrol and EVA4YOU Cohorts"

_nutrients, 2023, doi:10.3390/nu15245140_

Round 1

Reviewer 1 Report

Comments and Suggestions for Authors

Review of the manuscript “Determinants of diet quality in adolescents: results from the prospective population-based EVA-Tyrol and EVA4YOU cohorts”

I have read with interest the manuscript about diet quality and its determinants of Austrian adolescents aged 14 to 19 years that were included in two national prospective population-based cohorts. The diet quality was assessed with two scorings systems; AHEI-2010 and DASH indices. The authors revealed that those adolescents with higher AHEI-2010 and DASH scores were more likely to be female, had a higher level of physical activity and attended a profession-guided or general high school. Higher DASH scores were associated with older age and living in a high-income household. Determined nutrition knowledge was associated with a healthier diet assessed with AHEI-2010 and DASH scores, but participants’ beliefs about healthy and unhealthy diets and food weren’t associated with them. The authors also revealed that overall adolescents’ diet quality was rather low quality, with only 8% of those who met the recommended intake of vegetables, and a significantly high proportion of those who exceeded the recommended intake of sodium, and red and processed meat intake. The authors assessed adolescents’ socio-economic status which lower status was associated with lower diet quality when assessed with a DASH score. This study's results are in line with similar studies. The authors concluded that a nutrition education intervention program would be beneficial for this study's adolescent subgroups that had revealed determinants of lower diet quality.

Overall, the manuscript is written well, it presents valuable data in a simple and informative way regarding the studied aims, and it has a simply presented discussion and conclusion. Supplemental Material is also informative and can serve for further investigations.

Still, I have some suggestions for manuscript improvement.

Abstract: The first sentence should be rewritten, mentioning first that an unhealthy diet is a leading cause of death globally and then mentioning an early start in adolescence, since the next sentence presents the study aim of determinants of diet quality among adolescents.

Introduction: it is written well, but it lacks present knowledge about the connection between revealed determinants of unhealthy diet among adolescents, as well as their reasons and future implications.

Methods: The study population is well presented, but there could be more information about data collection and measurement procedures. If those are published, cite a reference, otherwise briefly mention tools and procedures for mentioned assessments.

Procedures and assessments: state which scale or scales were used for body weight measurements, describe with more detail the assessment of smoking status, which questionnaire was used for physical activity, and how the level of activity was calculated. For FFQ, what was the reported serving size? Did the questionnaire have examples of serving sizes listed, or did the adolescents enter their average portion by themselves? What period of time did the FFQ refer to, the last month, the last three months, or something else?

Statistical Analysis: was there a reason why the adjustments were not for smoking status and physical activity, since those factors may influence dietary intake and quality?

Results: the study results are presented clearly and succinctly. Table 1 could have p-values for a total sample’s differences between shown subgroups, and a p-value for differences between two groups of applied dietary scores.

Discussion: this section is presented clearly and shows relevant study results comparing them according to similar studies. Still, the authors could discuss about possible consequences of revealed lower diet quality, future possible implications on adolescents' health, possible solutions like the mentioned nutrition knowledge prevention program, what were the results of intervention programs aimed at adolescents' diet improvement like those the authors mention. When discussing those intervention programs, the sense of this study's results will be more accentuated, and more significant for local society. This discussion could actually put in process a needed nutrition intervention among subgroups of adolescents that had a lower diet quality. Also, there could be a brief discussion about this study's results revealing determinants of diet quality, why females and older adolescents had a better diet quality, and also why higher physical activity levels.

Conclusion: this section is well-written and has shown relevant results implications and possible solutions.

Comments on the Quality of English Language

A minor English language editing of some parts of the manuscript is suggested.

Reviewer 2 Report

Comments and Suggestions for Authors

The study has produced socially beneficial results by carefully analyzing valuable survey data and revealing factors affecting adolescents' dietary behaviors. However, the following points could be reconsidered.
1) The introduction section needed to be more extended and more balanced. The authors stated in the introduction: "However, to our knowledge, few studies have examined the prevalence of healthy eating using different diet quality indices in adolescents. However, the study should explain what issues need to be adequately addressed compared to previous studies on adolescent dietary behaviors and how the study attempted to address these issues using different dietary quality indices. The reviewer thinks the conclusions reached have already been pointed out in many previous studies.
(2) Table 4 shows the results of the regression analysis. When regression analysis is applied to the AHEI and DASH scores, the error terms in the respective regression equations may be correlated. If this is the case for this study, I think it is necessary to estimate the two regression equations simultaneously and correct the standard errors (i.e., the confidence intervals), assuming the error terms are correlated. Since there are few explanatory variables, the reviewer believes that it is highly likely that the error terms are correlated.
(3) Are many indicators added together? Even if the data are binary, such as the SES data, where the data are held or not held, principal component analysis based on the tetrachoric correlation matrix is possible, and it may be necessary to compute indicator values with a little more statistical processing.
